# Deep Convolutional Gated Recurrent Unit Combined with Attention Mechanism to Classify Pre-Ictal from Interictal EEG with Minimized Number of Channels

**DOI:** 10.3390/jpm12050763

**Published:** 2022-05-09

**Authors:** WooHyeok Choi, Min-Jee Kim, Mi-Sun Yum, Dong-Hwa Jeong

**Affiliations:** 1School of Computer Science and Information Engineering, The Catholic University of Korea, Seoul 14662, Korea; uoo1325@naver.com; 2Department of Pediatrics, Asan Medical Center Children’s Hospital, Ulsan University College of Medicine, Seoul 05505, Korea; pradoxwh@naver.com (M.-J.K.); yumyum99@hanmail.net (M.-S.Y.); 3Department of Artificial Intelligence, The Catholic University of Korea, Seoul 14662, Korea

**Keywords:** epilepsy, seizure, seizure prediction, ACGRU, channel-wise attention mechanism, 1D convolutional neural network, gated recurrent unit

## Abstract

The early prediction of epileptic seizures is important to provide appropriate treatment because it can notify clinicians in advance. Various EEG-based machine learning techniques have been used for automatic seizure classification based on subject-specific paradigms. However, because subject-specific models tend to perform poorly on new patient data, a generalized model with a cross-patient paradigm is necessary for building a robust seizure diagnosis system. In this study, we proposed a generalized model that combines one-dimensional convolutional layers (1D CNN), gated recurrent unit (GRU) layers, and attention mechanisms to classify preictal and interictal phases. When we trained this model with ten minutes of preictal data, the average accuracy over eight patients was 82.86%, with 80% sensitivity and 85.5% precision, outperforming other state-of-the-art models. In addition, we proposed a novel application of attention mechanisms for channel selection. The personalized model using three channels with the highest attention score from the generalized model performed better than when using the smallest attention score. Based on these results, we proposed a model for generalized seizure predictors and a seizure-monitoring system with a minimized number of EEG channels.

## 1. Introduction

Epilepsy is a neurological disorder in which transient dysfunction of the brain occurs chronically and repeatedly and can lead to seizures, loss of consciousness, numbness, and behavioral changes depending on the type of brain lesion. Epileptic seizures are characterized by excessive neuronal synchronization and these hypersynchronous electrical potentials can be recorded with scalp electroencephalography (EEG). Due to the spontaneous nature of epileptic seizures, continuous and long-term EEG recordings of patients are often needed for their accurate diagnosis and precise treatment [1,2]. However, the visual inspection of EEG signals is laborious, time-consuming, and subject to rater-biases even by expert neurologists. Therefore, there are numerous studies on automatic seizure detection to aid neurologists in the diagnosis of epilepsy and investigate the characteristics of epileptic signals.

Most studies on automatic seizure recognition have employed machine learning techniques to distinguish ictal EEGs from interictal or normal EEGs. Generally, EEG signals in epilepsy patients can be subdivided into the following four states: ictal, preictal, postictal, and interictal [3]. Ictal EEG activity can be observed during seizure onset. The preictal and postictal phases indicate periods immediately before and after a seizure, respectively. Interictal phases are non-seizure intervals between ictal phases. Most studies on seizure detection have focused on the classification of EEG from seizures from that of non-seizures. Various machine learning techniques that learn the epileptic patterns of temporal, spectral, and nonlinear features extracted from EEG signals, have been widely used for seizure detection. Machine learning classifiers including support vector machines, random forest, k-nearest neighbors, and artificial neural networks have resulted in a high classification accuracy of over 95% in seizure detection [4,5,6,7]. Although feature-based machine learning techniques have provided remarkable accuracy in seizure detection and the interpretability of epileptic signal properties, their performance could deteriorate owing to subject variability and noise from motion artifacts or the detachment of sensors.

Recently, deep learning techniques that learn epileptic characteristics from raw EEG signals without any feature extraction processing have shown great advances in seizure detection. Recurrent neural networks (RNN) and their variants such as long short-term memory (LSTM) and gated recurrent units (GRU) are frequently used for the analysis of time-series data, including EEG signals, because of their feedback loops. Many studies have proposed RNN-based structures for seizure detection by adopting multiple RNN, LSTM, or GRU layers [8,9,10,11,12,13,14]. Several studies have utilized CNN, which has been widely used in image recognition and seizure detection, and have successfully classified ictal and other states by converting EEG data into multi-dimensional tensors containing temporal, spatial, or spectral features [15,16]. A hybrid model that combines both CNN and RNN layers has also been proposed in several studies [17,18]. Despite the high performance in seizure detection, most studies have adopted within-patient protocols in which training and test datasets were collected from the same patient. Other studies evaluated model performance using record-wise cross-validation, whereby EEG signals from all subjects were randomly split into training and test sets. In other words, training and test sets can share data from the same subject. However, subject-specific or record-wise cross-validated models often result in the overfitting of data, which lowers the generalizability of the models. Since epileptic EEG signals can vary between patients due to different seizure types (focal vs. generalized) [19], age [20], sex [21], and brain lesions [22], the performance of automatic seizure detection models can worsen when applied to a new set of patients. Thus, subject-wise cross-validation where subjects in the training and test sets are independent must be performed to consider the between-subject variability in automatic seizure detection for clinical use.

To correct between-subject variability, some studies have adopted attention mechanisms in deep neural network models. The attention model, which captures the relevance between the encoder and decoder, was first introduced for machine translation [23]. Attention mechanisms that selectively focus on certain parts by learning and calculating the attention weights of all feature vectors have been applied in many tasks including text classification, machine translation, text summarization, syntax, and computer vision [24,25,26,27,28]. Channel-wise attention mechanisms have been adopted in EEG analyses because spatial information in EEG signals collected from multiple channels plays an important role in predicting brain status, such as emotion [29,30]. In seizure detection, because ictal signals are observed predominantly in channels near the focal area, the channel-wise attention mechanisms not only result in a high classification performance but also provide information on the contribution of each channel [31,32,33]. Furthermore, attention mechanisms have proved to be efficient in patient-independent seizure detection by exploring the significance of each channel in the classification of different patients [34,35].

Another challenge in the EEG-based diagnosis of epilepsy is the classification of seizures in the preictal stage. Several studies have reported high performance in the classification of ictal and interictal phases and also for ictal and preictal phases. Despite advances in anti-seizure medication and surgical intervention, 30% of patients with epilepsy are refractory to conventional intervention. The most devastating aspect of epilepsy is its “unpredictability” and the quality of life in these patients can be dramatically improved with accurate seizure forecasting in a pre-ictal state. Unfortunately, it is difficult to identify preictal patterns using long-term EEG signals because their EEG characteristics are not easily distinguishable from those in the interictal period [36]. In addition, although many studies have reported clinical findings of the preictal period, the exact timing of the preictal period has not been defined. Therefore, deep learning methods that autonomously capture characteristic patterns have been adopted to predict seizures from preictal EEG signals [8,9,10,11,12,13,14,15,16,17,18,19,20,21,22,23].

Most of the studies on the classification of epilepsy with EEG have aimed to distinguish seizure-related activity (EEGs from the ictal period) from non-seizure activity (EEGs from the interictal period or non-seizure patients) [8,14,15,16,17,33,34,35]. Although other studies have successfully predicted epilepsy using preictal EEGs, these models were built on the within-patient paradigm, implying that they are likely to perform poorly for a new set of data [37,38,39,40,41,42,43]. To utilize automatic seizure prediction in an actual clinical situation, a generalized model for cross-patient seizure prediction is required. To the best of our knowledge, there has been no study on generalized seizure prediction algorithms that can classify preictal from other phases via a between-patient approach. In this study, we aimed to classify seizures using preictal EEGs based on a hybrid model that combines a one-dimensional convolutional neural network (1D CNN) and a gated recurrent unit (GRU). We also blended an attention mechanism with the CNN-GRU model to overcome cross-patient variability problems. Furthermore, channel reduction through attention mechanisms would simplify the model for future applications in wearable patient monitoring systems.

Thus, the main contributions of this study, which aims to predict seizures using an attention-based 1D CNN coupled with GRU (ACGRU), are as follows:We propose a hybrid model that combines CNN and GRU layers to extract intrinsic temporal EEG patterns for differentiating preictal and interictal periods.We suggest that the attention mechanism in the encoder can successfully correct subject variability and that the self-attention mechanism in the decoder can capture informative temporal features.We propose the use of attention mechanisms as a novel means of channel selection for practical seizure prediction systems.

The remainder of this paper is organized as follows. In Section 2, an investigation of related studies on seizure detection and prediction using various approaches including conventional machine learning, deep learning, and attention mechanisms is presented. In Section 3, data collection methods and patient information are explained, and the detailed structure of the proposed ACGRU model is described in the remainder of the section. In Section 4, the classification results of the ACGRU and a model analysis are provided. Furthermore, in the last part of Section 4, an evaluation of the CGRU model with reduced channels using the attention mechanism is provided. Finally, in Section 5, we summarize and describe the main findings of our study.

## 2. Related Works

In this section, we examined the studies on the classification of seizure events. We divided the seizure studies into those that classified the ictal or preictal period from other periods. In Section 2.1, studies on seizure classification based on a deep learning approach are reviewed. In Section 2.2, studies on the attention mechanism, which has shown good performance in the interpatient model, are investigated. In Section 2.3, studies on seizure prediction, which distinguish EEGs of preictal states from those in other phases, are analyzed. In Table 1, we summarize the related studies on seizure detection with intra- or inter-patient paradigms and those on seizure prediction with intra-patient paradigms.

### 2.1. Studies on Deep Learning Based Seizure Detection

The most common type of deep neural network used for seizure detection is the RNN and its variants, which utilize their internal state as “memory” to process time-variant data. Yao et al. proposed an independent RNN (IndRNN) that uses 15-RNN layers to extract time-dependent features in a 23 s input sequence to classify seizures from non-seizures [8]. The model achieved 87% accuracy using cross-validation, outperforming convolutional neural networks (CNN) and LSTM. Vidyaratne et al. proposed a deep recurrent neural network (DRNN) that combines bidirectional RNN and cellular neural networks, which demonstrated 100% sensitivity with 7 s delays in a patient-specific experiment [9]. LSTM, which resolves the long-term dependency problem in conventional RNN by adopting the memory cells and gate mechanism, has successfully provided results with relatively lower computational costs than conventional RNN in many tasks dealing with time-series data [10]. Hu et al. proposed a deep bidirectional LSTM (bi-LSTM) network for automatic seizure detection [11]. To decrease the computational cost, statistical features extracted from the outputs of local mean decomposition were used as inputs for the bi-LSTM. As a result, a mean accuracy of 92.66% was achieved in the subject-specific experiment when using a 4 s window. Recently, gated recurrent units (GRU) have resulted in a similar performance to LSTM while simplifying the model structures and parameters [12]. Talathi designed a deep RNN with a GRU for the classification of healthy, interictal, and ictal EEG signals using a single channel [13]. A 98% accuracy in detecting seizure events was reported within 5 s using a 23.6 s window.

CNN, which is another popular deep learning method, was also used for seizure detection in several studies. Wei et al. presented a three-dimensional CNN (3D CNN) that converts EEG signals into 3D images to integrate multichannel information [14]. The 3D CNN successfully classified ictal signals with 93.83% accuracy and preictal signals with 92.57% accuracy in a 10-fold cross-validation test. Acharya et al. employed a 13-layer deep CNN and reported an accuracy of more than 90% for the differentiation between normal, preictal, and seizure periods using a 10-fold cross-validation strategy [15]. Zhou et al. reported that frequency-domain signals obtained from a fast Fourier transform were more suitable as inputs for CNN classifiers than as time-domain signals for seizure detection [16]. Moreover, a hybrid model that combines CNN and bi-LSTM layers was proposed for automatic epileptic seizure detection. The model resulted in 100% accuracy in the classification between normal and ictal cases using 10-fold cross-validation and a 23.6 s window [17]. Roy et al. proposed ChronoNet by stacking multiple 1D convolutional layers followed by GRU layers to classify normal and abnormal EEG data [18].

### 2.2. Studies on Attention Mechanism

In this section, studies that used the attention mechanism to classify EEG signals of various brain states, including seizures, are investigated. Yuan et al. adopted channel-aware attention mechanisms to identify the attentional representations of multichannel spectrograms. Their proposed method resulted in an average accuracy of 96.61% accuracy, outperforming other conventional machine learning methods for each patient. In addition, the attention scores were uniformly distributed within each patient [31]. Isaev et al. reported that attention mechanisms not only resulted in a 94.3% classification accuracy but also provided the importance level of each channel for seizure detection in neonatal patients [32]. Zhang et al. utilized attention models with a shallow VGGNet, which is a CNN-based model [33]. The proposed model used differential entropy as a feature, yielding an average accuracy of 82% in non-patient-specific k-fold cross-validation. Zhang et al. suggested that attention mechanisms combined with CNNs could provide an effective patient-independent diagnosis of seizures [34]. They demonstrated an average of 80.5% in the leave-one-out strategy, and the attention mechanisms showed the importance of each brain region in epileptic seizures based on its contribution to the classification. Yao et al. also proposed a deep learning approach that integrates the attention mechanism in capturing spatial features and BiLSTM to extract temporal features for the classification of seizures and non-seizures [35]. They reported that attention BiLSTM resulted in 87.8% accuracy with cross-validation and 83.89% accuracy with the cross-patient paradigm. Furthermore, they found that attention weights successfully spotlighted the target channels that reflected relatively larger ictal activities.

The attention mechanism combined with RNN has also been adopted in other more challenging EEG classifications, such as emotion recognition. Tao et al. proposed attention-based convolutional recurrent neural networks (ACRNN) to discriminate between arousal, valence, and dominance [29]. They demonstrated that the integration of channel-wise attention, convolutional, and recurrent layers led to higher performance in emotion recognition than when each layer was used alone. Chen et al. also utilized an attention mechanism with a hierarchical bidirectional GRU (H-ATT-BGRU) for emotion classification [30]. The proposed H-ATT-BGRU resulted in 67.9% accuracy in valence classification and 66.5% accuracy in arousal classification, outperforming other comparative models without the attention mechanism.

### 2.3. Studies on Seizure Prediction

Zhang et al. proposed CNN-based seizure prediction using a common spatial pattern [37]. They obtained 90% accuracy for the discrimination of the preictal and interictal states in leave-one-event-out cross-validation performed within each subject. Liu et al. utilized a multiview CNN to predict the occurrence of seizures using temporal and frequency features extracted with a deep canonical correlation analysis [38]. Khan et al. also adopted a CNN model to successfully extract preictal features from the 10 min period before seizure onset, suggesting that the interval could be referred to as the preictal period [39]. Tsiouris et al. employed an LSTM model using statistical features, spectral features, and connectivity measures for a seizure-prediction model [40]. In 10-fold cross-validation, by changing the window size from 15 min to 2 h, high performances with over 99% sensitivity were achieved in both patient-specific and global models. Daoud et al. combined semi-supervised learning based on a deep convolutional autoencoder (DCAE) with Bi-LSTM networks for patient-specific epileptic seizure prediction [41]. They were able to predict seizures one hour before onset with 99.6% accuracy in leave-one-event-out cross-validation. Usman et al. classified the preictal and interictal states using a three-layer CNN and LSTM [42]. They found that using generative adversarial networks (GAN) to resolve the data imbalance between preictal and interictal data by generating data samples of the minority class resulted in 93% sensitivity and 92.5% specificity with an average time of 32 min in k-fold cross-validation.

## 3. Materials and Methods

### 3.1. Patients and EEG Data

The EEG data of eight patients (three women and five men) among forty-one patients who had undergone epilepsy surgery at Asan Medical Center Children’s Hospital from July 2011 to July 2016 were chosen based on the following inclusion criteria. Surgical resection was guided by clinical factors, semiology, the visual assessment of long-term scalp video-EEG, Wada test, magnetic resonance imaging (MRI), single-photon emission computerized tomography (SPECT), and positron emission tomography (PET) results. Among those who had long-term video EEG and following surgical resection, patients who experienced two or more seizures during the monitoring period and those who were seizure-free for two years after surgery were included in this study. The data in this study were retrospectively analyzed; thus, the results of this study did not affect real-time surgical decision making. The average age of patients when EEG signals were recorded for this study was 18.25 ± 5.74 years old (min: 11, max: 26). The demographic and clinical data are provided in Table 2.

Scalp video-EEGs were recorded for three to five days with the TWin EEG system (Grass Technologies, West Warwick, RI, USA) using 32 channels according to the international 10–20 system. The sampling rate was 200 Hz and a 0.1 Hz high-pass filter was applied to the recording. Two epileptologists (M.K. and M.Y.) identified seizure onset based on the onset of clinical events and electrical changes by visual inspection. The onset of electrical seizures is characterized by sustained rhythmic discharges or repetitive spike-wave discharges that lead to habitual seizure semiology. Preictal data were defined as those data collected during the 10 min period immediately prior to the seizure onset without an overlap of previous ictal events. Interictal EEG data were collected from periods at least 12 h before or after the seizure. To minimize the influence of artifacts, inter-ictal EEGs were selected from the periods when patients were in resting states by visual inspection. The periods of eye movement, heart rate, and muscle artifacts were excluded. Pre-ictal EEGs were selected regardless of artifacts. In total, 670 min of preictal data from 67 seizure events and 27 h of interictal data were collected. The Institutional Review Board of the University of Ulsan College of Medicine, Seoul, Korea, reviewed and approved the study protocol (no. 2017-0074). Informed consent was waived due to the retrospective nature of the study.

In the analysis, we only used 18 common channels (Fp1, F3, C3, P3, F7, T7, P7, O1, Fp2, F4, C4, P4, F8, T8, P8, O2, Fz, and Cz) because of the different montages for each patient. EEG signals were spatially filtered using a common average reference to minimize bias regarding the distance from the referential electrodes. The irrelevant artifact was reduced using a 0.5 Hz high-pass filter and a 98 Hz low-pass filter. In addition, a 60 Hz notch filter was used to reduce AC power-line noise. EEG signals were segmented into 4 s epochs (800 data points) with a one-second sliding window. To reduce inter-trial variability, each epoch signal was normalized using a standard score.

### 3.2. Model Architecture

The proposed ACGRU model for predicting seizures in preictal EEGs includes the following five modules (Figure 1): (1) channel-wise attention module (Section 3.2.1), (2) convolutional module (Section 3.2.2), (3) GRU module (Section 3.2.3), and (4) self-attention module (Section 3.2.4). First, an attention mechanism was used to highlight channels containing relevant information for classification. Subsequently, three 1D CNN layers and four GRU layers were adopted to capture temporal features. Next, a self-attention mechanism encoded the temporal dependency within patterns extracted from the 1D CNN and GRU layers. Finally, in the classification layer, fully connected layers with a softmax function were used to distinguish the spatial and temporal preictal patterns from interictal patterns.

#### 3.2.1. Channel-Wise Attention Mechanism

As described in Section 2.1, EEGs were collected from 18 channels. Because EEG signals reflect the activity of brain regions beneath each channel, it is necessary to consider the contribution of each channel in the prediction of seizure occurrences as they vary based on patients, type of lesions, and phases. A channel-wise attention mechanism was employed to extract relevant features by assigning different weights to each channel to determine their significance. First, average pooling was performed throughout all time steps using preprocessed EEG signals that had undergone noise reduction and standardization. The attention matrix was then obtained via the FC layer and a nonlinear activation function using the channel-wise vector representing a mean value over time, which is as follows:(1)$$Y_{att}=softmax(W_{0}\cdot \left(\frac{1}{n}\sum_{1}^{n}X{)}^{T}+b_{0}\right)\;\;$$

In the above equation, the average pooling of EEG epochs X∈ℝn×c is used as an input and the attention matrix Yatt∈ℝ1×c is the final output in the attention mechanism, in which *n* is the number of time steps (i.e., 800 data points for 4 s epochs) and *c* is the number of channels (i.e., 18 channels). The weight matrix W0∈Rc×c and bias matrix b0∈ℝ1×c in the FC layers are updated during the training session. The *softmax* function is used for the activation function, which can be represented as follows:(2)softmax(vi)=exp(vi)∑j=1Cexp(vj)
where v=[v1, v2, …, vc] is the output of the FC layer. Finally, the output signal Xatt∈ℝn×c extracted from the channel-wise attention mechanism is obtained by multiplying the original EEG signal X and attention matrix Yatt as follows:(3)Xatt=X ⊗Yatt

The symbol ⊗ indicates element-wise multiplication (or the Hadamard product), which operates using the product of corresponding elements of two matrices when two matrices have the same dimension.

#### 3.2.2. Convolutional Neural Network Module

In general, 2D CNNs are often used in the field of computer vision because of their ability in adaptive learning of spatial features. In tasks using one-dimensional signals such as time series and text data, 1D CNNs can be applied to extract sequential contextual information. In this study, three 1D CNNs were utilized to extract temporal features from recalibrated EEG signals whose informative channels were highlighted using a channel-wise attention mechanism. The first convolutional layer performed the convolution operation on the recalibrated EEG signals Xatt∈ℝn×c, which is the output signal of the attention mechanism outlined in Section 3.2.1, in the direction of time step *n*. When performing convolutions with a kernel size of k, the number of output filters of f, and stride length of s by using the “same” padding (*p*) to maintain the input size, the output Yconv∈ℝ(n+2p−k)/s×f can be calculated as follows:(4)Yconv=∪i=1fELU(BN(Wi ∗ Xatt+bi)
where Wi∈ℝk×c and bi∈ℝ(n+2p−k)/s×1 refer to the convolutional kernel of i-th filter and the bias, respectively. The symbol * denotes the convolution operation along the temporal dimension and the *BN* function, which denotes batch normalization, was used for stable training. The batch normalization function *BN* can be described by the following equation:(5)BN(z)=zBN=γz−μBσB2+ϵ+β
where μB and σB are the mean and standard deviation of each batch, respectively, and γ and β are hyperparameters. The *ELU* function, which indicates an exponential linear unit, was used as the activation function. Finally, the output Yconv is obtained through concatenation, which is represented by the symbol ⋃, of the results from all f filters.

The remaining two convolutional layers successively performed the same operation based on Equations (4) and (5) with the same parameters as the first convolutional layers.

#### 3.2.3. Gated Recurrent Unit Module

GRU, which is known as an advanced model of RNN for solving long-term dependency problems by adopting a gating mechanism, was used to further extract temporal features from the output sequences of the convolutional module. Because the GRU utilizes an update and reset gate to yield a single hidden state, it is much simpler and faster than the LSTM model which uses an input, output, and forget gate to obtain a hidden and cell state (Figure 2). The update gate ut∈ℝ1×nh, which merges the functions of the forget and input gates in the LSTM model, determines the update ratio of past and present information, as described in the following equation:(6)ut=sigmoid(ht−1Wuh+ytWuy+bu)
where ht−1∈ℝ1×nh is the previous hidden state and yt∈ℝ1×f is the current input vector. The weight matrices Wuy∈ℝf×nh and Wuh∈ℝnh×nh and the bias bu∈ℝ1×nh of the update gate are set with the number of hidden units nh. In the first GRU layer, the input vector yt is the current output of the convolutional layer Yconv. Therefore, f is the number of output filters in the CNN. These values were consistent across all four GRU layers. The reset gate rt∈ℝ1×nh, which determines how much past information to retain or discard, can be obtained as follows:(7)rt=sigmoid(ht−1Wrh+ytWry+br)
where Wry∈ℝf×nh and Wrh∈ℝnh×nh are the weight matrices and br∈ℝ1×nh is the bias. The *sigmoid* function was used for the activation functions in both the update and reset gates.

Then, the candidate hidden state ht^ at time *t* can be obtained by performing element-wise multiplication (the symbol ⊗) between the previous hidden states and reset gate as follows:(8)ht^=tanh(ht−1Whh⊗rt+ytWyh)
where Whh∈ℝnh×nh and Wyh∈ℝf×nh refer to the weight matrices and tanh denotes the hyperbolic tangent function. Therefore, the previous hidden state is completely forgotten if rt=0.

Finally, the new hidden state $h_{t}$ can be calculated with the update gate $u_{t}$ where element-wise multiplication of the previous hidden state $h_{t-1}$ and current candidate state $\hat{h_{t}}$ is performed. The following equation shows the calculation for the new hidden states:(9)$$h_{t}=\left(1-u_{t}\right)⊗h_{t-1}+u_{t}⊗\hat{h_{t}}$$

In this study, four GRU layers were used to explore the temporal patterns in a sequence of CNN features. Because each GRU layer consisted of the same input and output sizes, all four GRU layers shared the same parameter sets.

#### 3.2.4. Self-Attention Mechanism and Classification

Because the temporal features extracted from the CNN and GRU are time-varying, the self-attention mechanism can help distinguish the contribution of each feature in seizure prediction. The self-attention mechanism was first suggested in machine translation to improve long-range dependencies [24]. In this study, we adopted multiplicative attention, which is known as Luong attention, to calculate the alignment function $a_{t}$ using the following equation:(10)$$a_{t}=\frac{exp\left(h_{t}{}^{T}h_{s}\right)}{\sum_{t}^{}exp\left(h_{t}{}^{T}h_{s}\right)}$$
where $h_{t}$ and $h_{s}$ refer to query and key, respectively. In this study, both query and key were set as the output state of the last GRU layer. Then, the new feature set $Y_{a}$ considering the context of the feature sequence $h_{t}$ can be represented using the attentive alignment score $a_{t}$ as follows:(11)$$Y_{a}=\sum_{t}^{}a_{t}h_{t}$$

Finally, classification between the preictal and interictal periods was performed using an FC with a *softmax* function. The probability of the classifier *P* can be calculated as follows:(12)$$P=softmax\left(WY_{a}+b\right)$$
where *W* and *b* are the weight and the bias, respectively. The model is trained by minimizing the cross-entropy loss between the predicted probability and the ground-truth label. The binary cross-entropy loss function is expressed as follows:(13)$$Loss=-\sum_{t}^{}\left(Y_{t}log\left(P_{t}\right)+\left(1-Y_{t}\right)log\left(1-Y_{t}\right)\right)$$
where $Y_{t}$ and $P_{t}$ represent the target label and predicted probability at time *t*, respectively.

### 3.3. Channel Minimization Using Attention Mechanism

To build an efficient seizure prediction system for patients with epilepsy, it is necessary to minimize the number of electrodes. Because the attention mechanisms in the first module indicate the importance of channels in seizure prediction, we hypothesized that attention weight computed in the ACGRU model can be used for channel selection. The attention layer resulted in different attention weights for each patient because of subject variability and owing to different seizure types or lesions. Therefore, the attention weight for each patient can be used to reduce the number of channels.

The proposed ACGRU model consists of a channel-wise attention layer, three CNN layers, four GRU layers, and a classification layer with a self-attention mechanism. While the channel-wise attention layer was used to extract spatial information that varied for each patient, the remaining layers were used to extract the temporal features within the EEG signals. Therefore, based on the collected attention score for each channel, CGRU classifiers were retrained using three channels with either the largest or smallest attention score. The performances of these retrained classifiers were compared to explore the potential use of the attention mechanism in channel selection. Because three channels with either the largest or smallest attention scores were selected, the channel-wise attention layer was removed in the CGRU model. Thus, the CGRU model, which consists of three CNN layers, four GRU layers, and a classification layer with the self-attention mechanism, was adopted for this step. The remaining settings and parameters were the same as those in the ACGRU model.

### 3.4. Hyperparameter Setting

When building deep CNN and GRU layers, it is important to optimize the hyperparameters because they determine the network effectiveness. The parameters were set based on previous studies and were tuned by examining the influence of each parameter on the validation performance [15,34,41]. As a result, three layers of the 1D CNN consisted of 32 filters with a kernel size of 4. The kernel was regularized with a penalty of 0.0001 to minimize overfitting. The “same” padding was used to retain the input size. However, to reduce the computational cost as well as to lessen the long-term dependencies problem that may occur in the next GRU layers, the number of strides was set to two in all three CNN layers. Thus, the input shape decreases by half every time it passes through each CNN layer. The four GRU layers had 32 hidden states with 50% dropout. The Adam optimizer was utilized, with an initial learning rate of 0.0001. When the maximum number of epochs was 30, the learning rate was updated to half of the initial learning rate after 20 epochs. Finally, the optimal batch size was set as 100.

## 4. Results

### 4.1. Subject-Wise Cross-Validation

To evaluate the performance of the proposed ACGRU model for seizure prediction, we adopted subject-wise cross-validation, where the EEG data of one subject were assigned to the test set, and those of the remaining subjects were employed to train the ACGRU model (Figure 3A). In the training session, we used the EEG data obtained up to 10 min before seizure onset for the preictal interval and employed them as the training set, similarly to previous studies [39,44,45]. Interictal EEGs with similar lengths to preictal EEGs were randomly selected. Therefore, 10 × *N* minutes of preictal and interictal data for seven patients were adopted when N seizure events were observed in those seven patients. The epochs were shuffled within the training set and divided into training and validation data at a ratio of 7:3.

To evaluate the trained ACGRU model, 10 min preictal data and whole interictal data from the remaining patient were used as a test set. This process was repeated eight times by substituting the patient into the test set. The following three metrics were adopted to evaluate classification performance: sensitivity, specificity, and accuracy. Sensitivity is defined as the number of true positives (TP), which is the number correctly classified as the preictal period, over the number of actual positive classes (i.e., all preictal periods including TP and false negatives (FN)). Thus, it evaluates the performance of the model in detecting preictal EEGs. Specificity is defined as the number of true negatives (TN), which is the number of accurately classified interictal periods, over the number of actual negative classes (i.e., all interictal periods including TN and false positives (FP)). Accuracy measures the model’s overall performance by calculating the ratio between the number of correct predictions and the total number of predictions including both preictal and interictal periods.

First, we evaluated the model performance using test data including EEGs from up to 10 min before seizure onset. The average accuracy of the classification of preictal and interictal EEGs using the ACGRU model was 82.86% (Table 3). The average sensitivity and specificity were 80% and 85.72%, respectively.

The average probabilities of all subjects obtained from the softmax layer, which was utilized in the classification layer to determine whether it belongs to the preictal phase, are illustrated in Figure 4. In this figure, probabilities near one and zero are classified as the preictal and interictal periods. The probabilities are consistently close to 1 throughout all of the 10 min periods.

### 4.2. The Influence of Preictal Period and Epoch Length

We examined how long the preictal period should be set to in order to train the seizure prediction models. Another ACGRU model was trained using preictal data obtained five minutes before seizure onset and interictal data with the corresponding data length. The evaluation was performed using 5 min preictal data and whole interictal data. The accuracy and probability of the softmax layer were decreased when the five-minute period before seizure onset was used for training. Using a five-minute preictal period, the sensitivity, specificity, and accuracy were 75.69%, 81.29%, and 79.75%, respectively. The average probability of a 5 min preictal period was 0.76, while that of a ten-minute preictal period was 0.83 (Figure 4).

In addition, the influence of epoch length on seizure prediction performance was further examined. To determine an optimal time window for seizure prediction, we tested the proposed ACGRU model using four different time windows as presented in Table 4. We determined that the accuracy and specificity were optimal when adopting a 4 s window, but the sensitivity was relatively lower than the specificity. The best sensitivity was observed when an 8 s window was utilized, but the difference was not significant. Meanwhile, the worst performance was observed when a 32 s window was utilized; thus, this result indicates that a longer epoch window did not guarantee higher classification performance.

### 4.3. Model Analysis

To examine the importance of each layer adopted in the proposed method, we compared the model performance by eliminating the attention mechanisms, convolutional layers, and GRU layers (Table 5). We determined that sensitivity, specificity, and accuracy largely decreased without attention mechanisms. In the CGRU model, the attention mechanism in the encoder which was utilized for capturing channel importance was removed, and the accuracy, specificity, and sensitivity decreased to 76.22%, 78.50%, and 73.94%, respectively. The self-attention mechanism adopted in the decoder, which measures the importance of extracted temporal features, was speculated to play a significant role in seizure prediction because the classification performance decreased without this layer. In addition, the mixture of 1D convolutional layers and GRU layers significantly affected the classification performance. The accuracy decreased to 55.17% without three 1D convolutional layers (AGRU) and decreased to 73.51% without four GRU layers (ACNN). In cases where a 1D CNN or a 1D GRU were separately adopted for the classification, the performance declined considerably.

We also compared the performance of other comparable models that resulted in high performance by adopting attention mechanisms or hybrid models combining CNN and RNN in seizure detection or emotion recognition. The AbiLSTM, which performed well in seizure detection for the cross-patient experiment, comprised the attention layers for channels followed by biLSTM [35]. Thus, it substituted the 1D-CNN and GRU layers with LSTM and excluded the self-attention mechanism. Chorononet, which performed well at representing patterns of time series data including EEG and speech, comprised three 1D convolutional layers and four GRU layers [18]. Therefore, it excluded both attention mechanisms. The ACRNN, which provided good performance in EEG-based emotion classification, comprised a channel-wise attention mechanism, CNN, two-layer LSTM, and a self-attention mechanism [29]. Its convolutional layer functioned channel-wise, whereas the 1D convolutional layers of ACGRU functioned to capture temporal patterns. The proposed ACGRU model outperformed the other three methods. The performance of the ACRNN model that adopts channel-wise attention and channel-wise convolutional layers with LSTM and self-attention was the best among those models, except for the ACGRU model. For a comparison of the classification performance, the same training and test dataset as the ACGRU model were used for each of these models.

### 4.4. Attention Mechanism-Based Channel Reduction

The attention mechanism assigns a weight to each channel, and this indicates the importance of each channel in the seizure prediction. To assess whether the channels with high attention scores were more informative for seizure prediction, the CGRU models were trained with a reduced number of channels using the attention score. As presented in Section 4.1 and Section 4.2, the ACGRU model was evaluated using leave-one patient-out cross-validation that evaluated one patient’s test set separately from the training set of the remaining patients. Therefore, this approach cannot be applied to the CGRU model, because it needs to select relevant channel sets with high attentive scores that are different for each patient. Accordingly, the CGRU model with attention-based channel selection was evaluated for each subject (Figure 3B). First, when the number of seizure events was *N* for each patient, we split the EEG data into the preictal EEG of one test event and *N* − 1 training events. Interictal EEG signals were randomly divided into training and test sets according to the length of the preictal EEG. In addition, the attention score for each channel in *N* − 1 training sets was calculated using the attention weight of the ACGRU model trained with the other subjects. Furthermore, the CGRU model was trained for each patient by adopting EEG data corresponding to the reduced channels in *N* − 1 training sets. Finally, the test EEG dataset, which was not adopted to train the CGRU model, was used to evaluate the performance of the CGRU model with attention-based channel selection. This process was repeated *N* times by permuting the training and testing sets.

The two different CGRU models were trained by selecting the top three or six channels with the highest attention scores and the bottom three or six channels with the lowest attention scores. The learning rate was altered from 0.0001 to 0.001, owing to the decreased number of data because the experiment was performed with a reduced number of channels within each patient. The classification performance with the top three channels was significantly higher than that with the bottom three channels. The top three channels resulted in 80.86% accuracy, 75.03% sensitivity, and 86.69% specificity, whereas the bottom three channels resulted in 75.35% accuracy, 55.63% recall, and 95.07% specificity (Figure 5). When the top six channels were selected, the average accuracy was 87.58%, while that with the bottom six channels was 82.13%. Therefore, these experimental results indicate that the attention score adopted in ACGRU could offer discriminative spatial information.

When observing channel selection using the attention score, we determined that the selected channels in all folds were consistent within each subject (Table 6). Compared to the EEG channels related to the epileptogenic foci, the top three channels did not overlap, instead, they were observed on the opposite side of the epileptogenic foci.

## 5. Discussion

In this study, we proposed a deep neural network including 1D CNN and GRU layers with an attention mechanism for seizure prediction that can distinguish epileptic seizures from EEG signals during the preictal period. Because it is difficult to determine distinct epileptic patterns in preictal EEGs compared to ictal EEGs, conventional feature-based machine learning methods, which exhibit great performance in seizure detection, have limitations in seizure prediction. The proposed method successfully identified seizures in preictal EEGs because the 1D CNN and GRU layers can autonomously capture intrinsic temporal patterns in EEG signals. Moreover, the channel-wise attention mechanism, which provides higher weight for informative channels, played an important role in calibrating patient variability that occurs owing to the different types and lesions of each patient. The proposed ACGRU model resulted in an 82.86% accuracy, which outperformed other deep learning models that have reported high performance in seizure detection (Table 5). The sensitivity, which is the ability to correctly classify the preictal period, was 80% on average. For Patients 1, 4, and 7, the sensitivities, were 66.67%, 70.90%, and 65.38%, respectively, were much lower than those of the other patients. After classification, additional visual inspection of raw EEG signals was carried out, which showed that those of Patients 1, 4, and 7 had a relatively low amplitude with fast activities (Appendix A). The specificity can be defined as the ability to correctly identify the interictal period; thus it indicates the ability to prevent false alarms. The average specificity was 85.72% even with long-term interictal data for a duration of 2 to 5 h for each patient. We did not use precision, which is defined as the ratio between the number of TP and that of all labels predicted as positives (TP + FP), to evaluate model performances. Since the total periods of interictal and preictal data observation were 27 h and 11.17 h (670 min), the number of FP was much larger than that of TP despite the high specificity.

To evaluate the contribution of each layer to the seizure prediction, we compared the accuracy when each module was excluded (Table 5). When channel-wise attention was removed, the accuracy decreased by approximately 6%. There was a larger decline in the accuracy when both channel-wise attention and self-attention mechanisms were removed. When adopting either one of the CNN and GRU modules, a large decrease in the accuracy was observed. In particular, when we excluded 1D convolutional layers, the accuracy decreased by approximately 27%.

To further examine the influence of the channel-wise attention mechanism on seizure prediction, we retrained the CGRU model with a reduced number of channels using the attention score. The performance of the CGRU model trained with the top three or six channels with the highest attention score was better than when the model was trained with the bottom three or six channels (Figure 5). In the CGRU model, using the top three channels resulted in 80.86% accuracy and 75.03% sensitivity while using the bottom three channels resulted in 75.35% accuracy with 55.63% sensitivity. This result indicates that the channel-wise attention mechanism successfully captured discriminative spatial information for seizure prediction. In addition, we suggest that the channel-wise attention mechanism can be utilized as a channel selection method. We observed that the top three or six channels selected by the average attention score did not differ between trials in each patient (Table 6). Thus, we suppose that this mechanism can determine common spatial patterns of preictal activities in each patient, although individual attention scores were slightly different between each epoch. We also determined that the channels selected using channel-wise attention were not always equivalent to the channels related to the epileptogenic foci identified by the neurologist. Interestingly, the channels with the largest attention score were more often found in the hemisphere opposite the lesion. D’Alessandro et al. also reported that the best channel of intracranial EEG (iEEG) for seizure prediction was contralateral to the focus when using feature selection based on a genetic algorithm [44,45]. In other EEG studies, preictal desynchronization was determined in the hemisphere contralateral to the focal area [46]. Therefore, when using on-scalp EEG for the seizure prediction, the use of channels in both hemispheres rather than those lateralized to seizure foci could lead to higher performance. Moreover, contralateral channels could be considered when the number of EEG channels is reduced for seizure monitoring.

The accuracy of the patient-specific CGRU model was relatively low for Patient 8 compared to other patients (Appendix B). This could be due to the scarcity of preictal data because this patient experienced only four seizures during EEG monitoring. In addition, Patient 1 exhibited low performance in seizure prediction due to low sensitivity, which was consistent with the low performance of the generalized ACGRU model. This could be because the relevant ROIs are larger or because EEGs were low in voltage for Patient 1 compared to the other patients (Appendix A). Although normalization was executed during the deep learning process, such a difference could deteriorate the classification performance.

In this study, the preictal period utilized for training was set to 10 min before seizure onset. In the evaluation of ACGRU models presented in Section 4.1, we demonstrated that the probabilities of the softmax layer were larger than 0.8 in most epochs (Figure 4). Although training with the 10 min period was satisfactory for seizure prediction, which outperformed the result based on the 5 min preictal period, it is important to determine the optimal time onsets to define the preictal period. Several studies have adopted the preictal period differently, such as 5/10/30 min before seizure onset [37,39,44,45,47,48]. However, there was no consensus regarding the preictal period because of the limitations of preictal-related biomarkers. Because the proposed model extracted the intrinsic temporal patterns related to the preictal period, we expect to increase the model performance and find the optimal preictal period and related biomarkers by collecting long-term epileptic EEGs that are much longer than 10 min in the future.

Based on our results, despite the limitations, we suppose that the ACGRU and CGRU models could be employed for seizure monitoring in the following clinical scenarios. First, when a new inpatient with epilepsy enrolls, 18 channels of EEG electrodes can be attached according to the 10–20 international system. Then, seizure events will be detected in advance using the ACGRU trained with previously acquired EEG data from other patients. Long-term EEGs can be collected during the patient’s hospital stay, and then the channel can be reduced using the attention score of the collected EEG. When the patient leaves the hospital, personalized devices with several electrodes selected from the attention score can be provided to each patient to monitor seizure occurrences. Additional experiments with larger datasets covering various cases need to be performed for the clinical employment of the ACGRU model. For instance, this model can not only be utilized to distinguish psychogenic nonepileptic seizures (PNES), which lead to epilepsy-like behaviors without ictal EEG changes, but also to find PNES biomarkers [49]. In conclusion, this study proposes a novel deep ACGRU model, which can extract intrinsic temporal preictal patterns and investigate the related ROIs, that can overcome the current limitations in seizure prediction using the cross-subject paradigm. In addition, the proposed methods can effectively reduce the number of EEG electrodes required for personalized seizure monitoring.

## Figures and Tables

**Figure 1 jpm-12-00763-f001:**
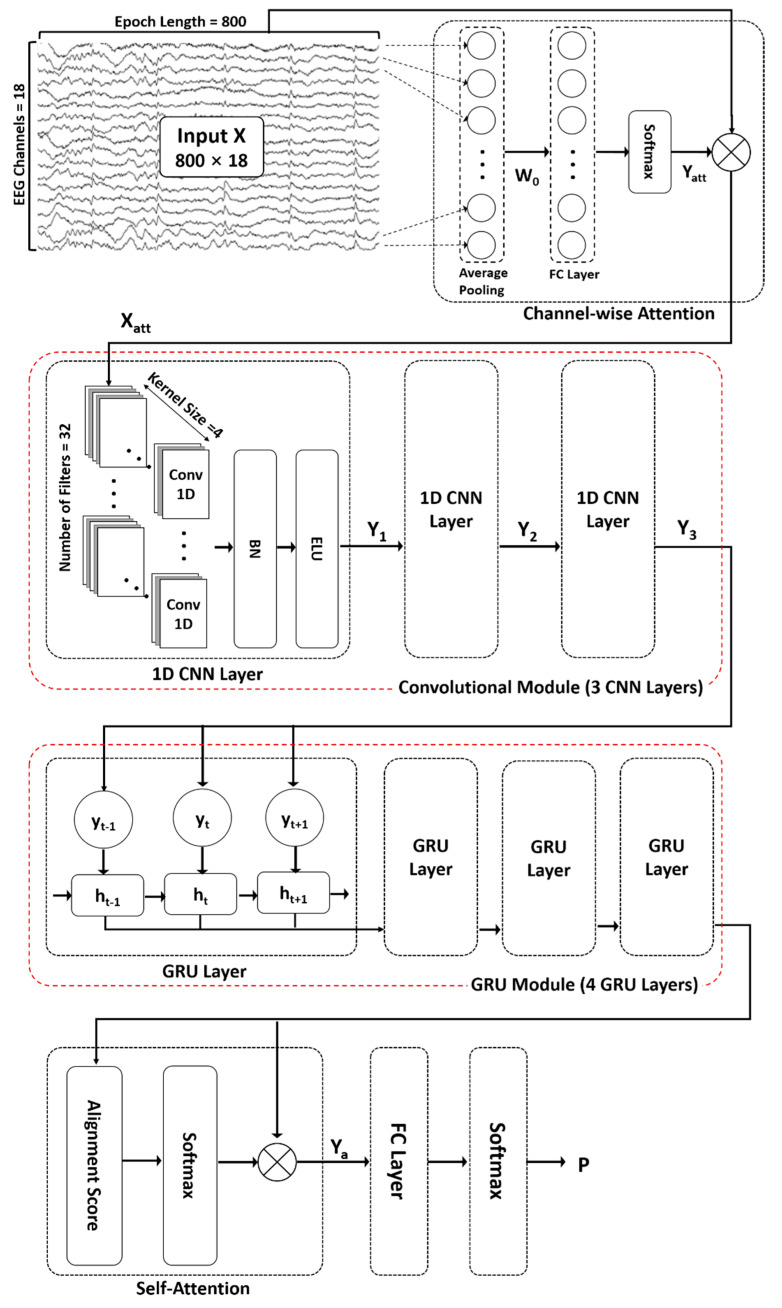
The proposed attention-based convolutional gated recurrent unit model (ACGRU). The model includes three one-dimensional convolutional layers (Conv1D) and four gated recurrent units (GRUs) combined with channel-wise attention mechanisms to capture temporal and spatial information of a four-second input epoch (800 data points × 18 channels). Captured patterns were used for the classification between the preictal and interictal periods using self-attention mechanisms and fully connected layers (FC) with softmax function. ⊗: element-wise multiplication, BN: Batch normalization, ELU: exponential linear unit.

**Figure 2 jpm-12-00763-f002:**
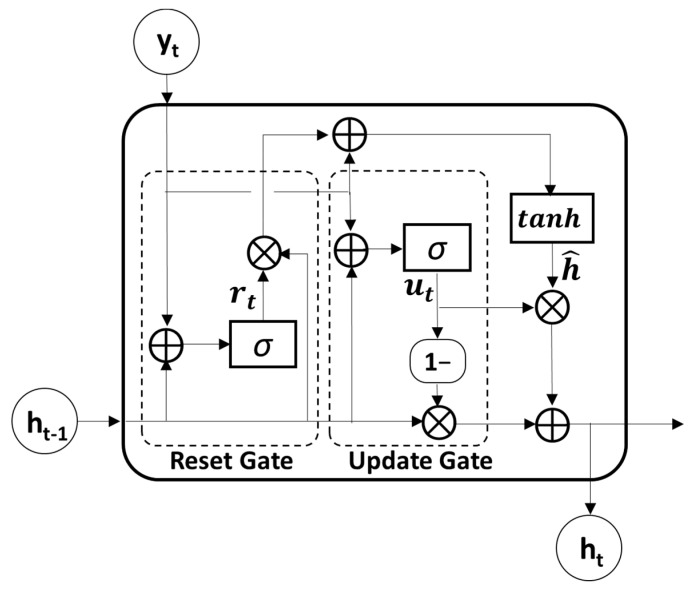
The structure of gated recurrent unit (GRU). The symbol ⊗: the element-wise multiplication, ⊕: vector summation, *σ*: sigmoid function, tanh: hyperbolic tangent function.

**Figure 3 jpm-12-00763-f003:**
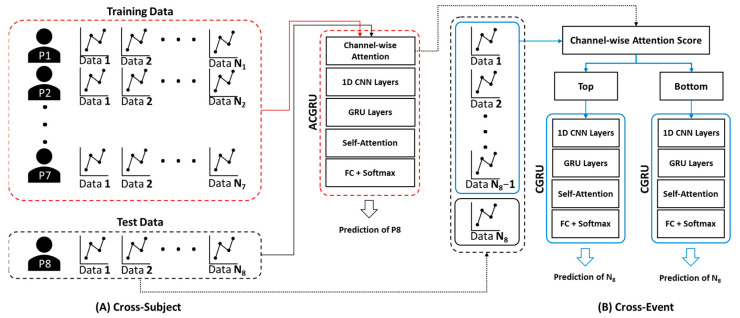
The method for cross-validation of (**A**) ACGRU and (**B**) CGRU models. (**A**) The ACGRU models aimed to classify preictal seizures using a between-subject experiment, which trained with seven patients and tested with the remaining one patient. The process was repeated eight times by permuting the test data. (**B**) The CGRU models were retrained with the top or the bottom three (or six) channels selected based on the attention score of each individual patient from the pretrained ACGRU model. The *N* − 1 preictal data were adopted for training and the remaining one preictal data were used as the test data when *N* seizures were detected in a patient. The process was repeated *N* times by permuting the test data.

**Figure 4 jpm-12-00763-f004:**
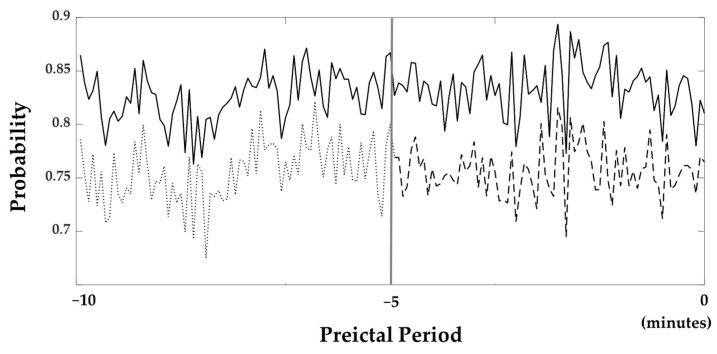
Probabilities resulted from the softmax layer. The bold line is the probability when the 10 min preictal period was used as the training set. The dashed line is the probability when the 5 min preictal period was used. Furthermore, the probability 5 to 10 min prior to seizure onset was also shown with the dotted line. The gray line located in the middle of the figure indicates 5 min prior to seizure onset.

**Figure 5 jpm-12-00763-f005:**
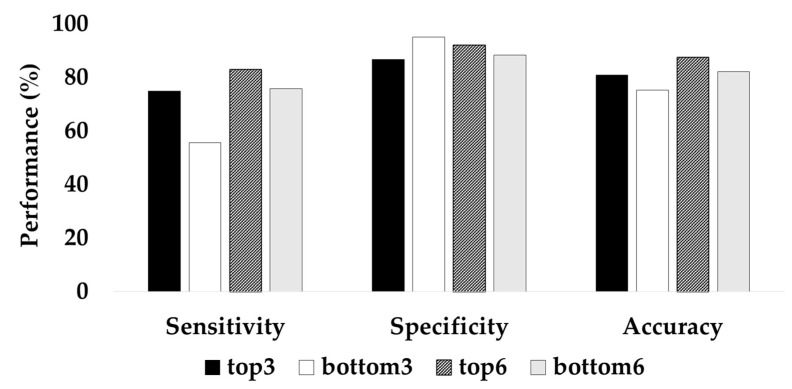
Classification accuracy of the CGRU model with reduced channels based on the attention score.

**Table 1 jpm-12-00763-t001:** Summary of related studies.

Reference	Accu. (%)	Epoch (s)	Feature Extraction	Model	Class
Seizure Detection (Intra-Patient)					
Vidyaratne [9]	100	1	Raw EEG	RNN	ictal vs. interictal
Hu [11]	93.61	4	Local Mean Decomposition	BiLSTM	ictal vs. interictal
Talathi [13]	98	23.6	Raw EEG	GRU	ictal vs. interictal vs. normal
Zhou [16]	97.5	1	Spectral Features	CNN	ictal vs. interictal
Yuan [31]	96.61	-	Spectral Features	Attention	ictal vs. interictal
Seizure Detection (Inter-Patient)					
Yao [8]	87	23	Raw EEG	RNN	ictal vs. interictal
Wei [14]	92.38	10	Raw EEG	3D CNN	ictal vs. preictal vs. normal
Acharya [15]	88.67	23.6	Raw EEG	CNN	ictal vs. interictal
Abdelhamed [17]	98.89	23.6	Raw EEG	CNN + BiLSTM	ictal vs. interictal vs. normal
Zhang [33]	82	1	Differential Entropy	VGGNets + Attention	ictal vs. interictal
Zhang [34]	80.5	1	Raw EEG	CNN + Attention	ictal vs. interictal
Yao [35]	83.72	23	Raw EEG	BiLSTM + Attention	ictal vs. interictal
Seizure Prediction (Intra-Patient)					
Zhang [37]	90	5	Common Spatial Pattern	CNN	preictal vs. interictal
Liu [38]	85.5	30	Spectral Features	Multi-view CNN	preictal vs. interictal
Khan [39]	87.8	1	WT	CNN	preictal vs. interictal
Tsiouris [40]	99.28	5	Correlation, Temporal & Spectral features	LSTM	preictal vs. interictal
Daoud [41]	99.6	5	Raw EEG	CNN + Autoencoder + BiLSTM	preictal vs. interictal
Usman [42]	93	29	EMD, STFT	CNN+LSTM	preictal vs. interictal
Seizure Prediction (Inter-Patient)					
No study was reported.					

**Table 2 jpm-12-00763-t002:** Information on patients with epileptic seizure.

Patient	Age	Sex	# of Seizure	Length of Interictal EEG	Epilepsy Type/Epileptogenic Lesion	Related Channels
1	12	M	7	2 h	DNET1 + FCD2/Right temporal	Fp2, F8, T8, P8, F4, C4, P4
2	13	M	7	3 h	FCD/Right frontotemporal	F8, T8, F4, C4
3	19	M	10	5 h	Subpial Gliosis + HS3/Right temporal	F8, T8, P8
4	16	M	10	3 h	Subpial Gliosis + HS/Right temporal	T8
5	25	F	8	4 h	FCD + HS/Right temporal	T8
6	26	F	13	3 h	HS/Right temporal	T8, P8
7	11	F	8	3 h	DNET/Right frontotemporal	F8, T8, F4, C4
8	24	M	4	4 h	HS + Cavernous hemangioma/left temporooccipital	T7, P7, O1

DNET: Dysembryoplastic Neuroepithelial Tumor, FCD: Focal Cortical Dysplasia, HS: Hippocampal Sclerosis.

**Table 3 jpm-12-00763-t003:** The classification results of the ACGRU model for each subject.

Patient	Sensitivity	Specificity	Accuracy
1	66.67%	94.00%	80.33%
2	91.81%	78.38%	85.10%
3	90.43%	86.43%	88.43%
4	70.90%	91.47%	81.18%
5	81.04%	95.00%	88.02%
6	94.10%	83.18%	88.64%
7	65.38%	86.48%	75.93%
8	79.67%	70.83%	75.25%
Average	80.00%	85.72%	82.86%

**Table 4 jpm-12-00763-t004:** Accuracy depending on the epoch length.

	Length	Sensitivity	Specificity	Accuracy
Epoch Window	4 s	80.00%	82.72%	82.86%
	8 s	82.43%	80.58%	81.50%
	16 s	79.31%	77.67%	78.49%
	32 s	70.88%	76.27%	73.58%

**Table 5 jpm-12-00763-t005:** The comparison of classification performance between the proposed ACGRU model, ablation model, and the models employed in other studies. Our dataset was used for each model.

	Model	Sensitivity	Specificity	Accuracy
Ablations	CNN only	72.94%	73.71%	73.32%
	GRU only	55.98%	79.73%	67.85%
	CGRU (CNN + GRU)	71.54%	71.29%	71.41%
	AGRU (Attention + GRU)	55.85%	54.48%	55.17%
	ACNN (Attention + CNN)	74.05%	72.97%	73.51%
	Removal of Channel-wise Attention	73.94%	78.50%	76.22%
	Removal of Self-attention	74.83%	76.45%	75.64%
Other Comparative Models	Attention LSTM [35]	74.39%	72.41%	73.40%
	Chorononet (CGRU) [18]	71.21%	76.47%	74.09%
	ACRNN [29]	76.47%	82.57%	79.52%
Proposed ACGRU Method		80.00%	82.72%	82.86%

**Table 6 jpm-12-00763-t006:** Channels selected for each individual patient based on the attention score. The symbol * denotes channels contralateral to a region of interest (ROI) related to epileptogenic foci. The symbol ^†^ indicates channels corresponding to ROI.

Patient	1	2	3	4	5	6	7	8
Top3	F7 *	Fz	F3 *	Fz	Fz	C3 *	F3 *	F3
	O2	P7 *	Fz	P7 *	Fp1 *	O1 *	O1 *	F4 *
	P3 *	T7 *	Fp1 *	T7 *	T7 *	O2	T7 *	P4 *
Bottom3	F3 *	C3 *	Fp2	Cz	C3 *	F4	C3 *	Fp2 *
	F8 ^†^	Cz	O1 *	Fp2	F7 *	FZ	C4	Fp1
	P7 *	Fp2	O2	Fp1 *	P7 *	P8 ^†^	P3 *	P8 *
Top6	Cz	F8 ^†^	F3 *	F7 *	C4	C3	CZ	C4 *
	F7 *	Fz	F4	Fz	Fz	CZ	F3 *	F3
	Fp2 ^†^	P4	F8 ^†^	O1 *	Fp1 *	O1	F7 *	F4 *
	O2	P7 *	Fz	P3 *	O2	O2	Fp1 *	O2 *
	P3 *	P8	Fp1 *	P7 *	P8	P3	O1 *	P4 *
	T7 *	T7 *	T8 ^†^	T7 *	T7 *	P4	T7 *	T8 *
Bottom6	C4 ^†^	C3 *	F7 *	C3 *	C3 *	F4	C3 *	F8 *
	F3 *	C4 ^†^	Fp2	Cz	Cz	F7 *	C4 ^†^	Fp2 *
	F8 ^†^	Cz	O1 *	Fp2	F3 *	F8	O2	Fp1
	Fp1 *	Fp2	O2	Fp1 *	F7 *	Fz	P3 *	O1 ^†^
	P4 ^†^	P3 *	P7 *	O2	P3 *	Fp1 *	P7 *	P3
	P7 *	T8 ^†^	P8 ^†^	P4	P7 *	P8 ^†^	P8	P8 *
ROI	Fp2	F8	F8	T8	T8	T8	F8	T7
	F8	T8	T8			P8	T8	P7
	T8	F4	P8				F4	O1
	P8	C4					C4	
	F4							
	C4							
	P4							

## Data Availability

The data can be provided after the approval of the Institutional Review Board of Asan Medical Center. Please contact the author.

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
