# Peer review of "Deep Convolutional Gated Recurrent Unit Combined with Attention Mechanism to Classify Pre-Ictal from Interictal EEG with Minimized Number of Channels"

_jpm, 2022, doi:10.3390/jpm12050763_

Round 1

Reviewer 1 Report

The manuscript is devoted to the application of modern data mining and machine learning techniques to analyze the interictal EEG signal in order to classify preictal and interictal phases of epileptic seizures. The topic is very interesting and actual. The EEG signal is very difficult to achieve high classification accuracy.  The authors have proposed a hybrid model that combines one-dimensional convolutional layers (1D CNN), gated recurrent unit (GRU) layers, and attention mechanisms to classify preictal and interictal phases. The introduction section and literature review were written correctly. The authors have analyzed the current stage of the research in this subject area. Their analysis is very interesting. The theoretical part also has written clearly. Results and discussion allow us to understand the contribution of the research.

However, to my best mind, there are some positions, which should be explained in more detail. Below, I present my remarks.

  1. The CNN network effectiveness depends on the values of hyperparameters. I think that the paper will look better if the authors explain in more detail their choice. For example, the length of the filter, number of kernels, etc. Did you optimize these parameters?
  2. As I see, we used 1D two levels CNN. Please, explain your choice.
  3. Did you consider the length of the signal when you set up the length of the filter? How do you perform the padding step? Please, explain this.

In general, work is very interesting. I think that the manuscript can be accepted after minor revision. 

Author Response

The authors thank you for reviewing the manuscript entitled “Deep Convolutional Gated Recurrent Unit Combined with Attention Mechanism to Classify Pre-ictal from Interictal EEG with Minimized Number of Channels” and for your valuable comments and suggestions to improve the manuscript.

Reviewer 2 Report

In the paper titled “Deep Convolutional Gated Recurrent Unit Combined with Attention Mechanism to Classify Pre-ictal from Interictal EEG with Minimized Number of Channels”, the authors proposed a deep neural network including 1D CNN and GRU layers with an attention mechanism for seizure prediction that can distinguish epileptic seizures from EEG signals during the preictal period. 
The manuscript addresses an important topic in health care, one of the most impacting neurological disorders worldwide. The scientific sound of this paper is interesting from my point of view.  I would like to suggest to the authors a few tips:

  1. The first section, the introduction, is too long. I would recommend careful review, refocused, and pruning it.
  2.  In the first section I found “u-health”. I was wondering about the meaning.
  3.  In the first section I would recommend “A comprehensive machine-learning-based software pipeline to classify EEG signals: A case study on PNES vs. control subjects”
  4. I suggest refocusing and shortening the related works section. How it is, is too long and confusing.
  5.  In section 3.1 “Among patients who have experienced two or more seizures during the monitoring period, eight of them who have been seizure-free for two years after surgery were included for this study. The data in this study were retrospectively analyzed; thus, the results of this study did not affect the real-time surgical decision making.” Is so confusing to me.
  6. I would suggest reporting demography’s information as mean+/-std.
  7.  In section 3.1 the following part “Scalp video-EEG were recorded for three to five days with the TWin EEG system (Grass Technologies, West Warwick, Rhode Island, USA) using 32 channels according to the international 10–20 system. The sampling rate was 200 Hz and a 0.1 Hz high-pass filter was applied to the recording. Two epileptologists (M.K. and M.Y.) visually inspected the video-EEG and identified seizures based on the initial clinical and electrical change. The initial electrical change is defined as sustained rhythmic discharges or repetitive spike-wave discharges that lead to habitual seizure symptoms. We treated 10 minutes immediately before seizure onset without any ictal period as preictal data. Interictal EEGs were collected from periods at least 12 hours before or after the seizure. In total, 670 minutes of preictal data from 67 seizure events and 27 hours of interictal data were collected. The Institutional Review Board of the University of Ulsan College of Medicine, Seoul, Korea, reviewed and approved the study protocol (no. 2017-0074). Informed consent was waived due to the retrospective nature of the study by the Institutional Review Board of the University of Ulsan College of Medicine.” Is confusing.
  8.  In section 3.1 how does the author clean data from, eyes movement, heart rate, muscle artifact, etc?
  9. In section 3.1 “Each epoch signal was normalized using a standard scaler.” I would like to ask about the meaning of this sentence.
  10. In Figure 1 I would like to suggest adding some input dataset shape and size, layers size, and so on.
  11. The “symbol ⊗ indicates the element-wise multiplication” is redundant in this manuscript. I would suggest defining it once a time. 
  12.  In section 3.2.2. Convolutional Neural Network Module, what does “recalibrated EEG” mean?
  13. The equations used to describe the deep models have been already well addressed in past literature, thus I was wondering if it is more useful to reference Convolutional Neural Network Module and Gated Recurrent Unit Module equations and spend more time describing the used models. More info about the models may be important for reproducibility.
  14.  I would suggest splitting Section 4.1  to highlight another important experiment, the examination of epoch length to increase classification accuracy.
  15.  I suggest checking the English language, redundancy, and bad sound in the sentences. 

Author Response

(The authors gave the same response as above.)

Round 2

Reviewer 2 Report

In my opinion the authors did a great job improving the manuscript.